# Suffering and Healing in the Context of LVAD Treatment

**DOI:** 10.3390/jcm8050660

**Published:** 2019-05-11

**Authors:** Kristin Kostick, Meredith Trejo, J.S. Blumenthal-Barby

**Affiliations:** Baylor College of Medicine, Center for Medical Ethics and Health Policy, Houston, TX 77030, USA; Meredith.Trejo@bcm.edu (M.T.); jennifer.blumenthal-barby@bcm.edu (J.S.B.-B.)

**Keywords:** illness narratives, LVAD, patient experience, suffering and healing

## Abstract

Background: Illness narratives with meaningful, competent and targeted content have been shown to provide useful guides for patient decision-making and have positive influences on health behaviors. The use of narratives in decision aids can confer a sense of structure, plot and context to illness experiences and help patients make treatment decisions that feel sensible, informed, and transparent. Aim: This paper presents narratives of suffering and healing from patients and their caregivers with advanced heart failure who engaged in decision-making regarding Left Ventricular Device Assist (LVAD) treatment. Methods: Narratives were collected from in-depth interviews with patients who accepted (*n* = 15) versus declined (*n* = 15) LVAD implant, LVAD candidates who had received education about LVAD and were in the process of making a decision (*n* = 15), and caregivers (family or significant others) of LVAD patients (*n* = 15). Results: Participants shared “restitution” narratives that most commonly conveyed a shift from pre-implant physical suffering and “daily hell,” fatigue so intense it “hurts,” along with emotional suffering from inability to engage with the world, to post-implant improvements in mobility and quality of life, including positivity and family support, adaptation on a “journey,” “getting one’s life back” and becoming “normal” again. Conclusion: For LVAD patients, other patients’ illness narratives can help to give meaning to their own illness and treatment experiences and to more accurately forecast treatment impacts on lifestyle and identity. For clinicians, patient narratives can enhance patient–practitioner communication and understanding by highlighting perspectives and values that structure patients’ clinical experiences.

## 1. Introduction

Facing a significant illness event has been likened to “ontological death,” or a collapse of meaning and identity, whereby interpretations of the self, including one’s capabilities, ambitions and future, are reconsidered and reconstituted [1].

Anecdotal evidence suggests that the biomedical tendency to objectify illness as a “thing” rather than as an integral part of a person’s being and experience can unintentionally leave patients without the tools to cope with the psychological and existential crises often involved in facing illness and recovery [2]. This “crumbling away of [the] former self-image without simultaneous development of equally valued new ones,” as one medical sociologist characterized it, can lead to significant distress for patients and caregivers alike [3] (p. 168).

One way to improve coping with severe illness may be to provide patients with narratives that describe similar illness experiences from other patient perspectives. Narratives with meaningful, competent [4] and targeted [5] content have been shown to provide useful guides for patient decision-making [6,7,8] and to have positive influences on health behaviors [9]. Illness narratives help to open frameworks for individuals to discover significance in their experiences, as well as to renegotiate new identities, capabilities and goals in personally satisfying ways [1]. Research on the use of narratives suggests that conveying illness experiences through a story timeline can confer a sense of structure, plot and context to help patients make decisions that feel sensible, informed, and transparent. As Dohan et al. [10] phrased it in their explanation of the importance of integrating narrative into health care decision-making, “Narrative story lines don’t tell patients or clinicians what to do so much as remind them of the pathways and choices they face.” The purpose of a patient narrative is not to arrive at a specific “ending,” but to provide a means by which patients can negotiate and renegotiate the various meanings and significance of illness [11].

This paper presents narratives from patients (and their caregivers) with advanced heart failure who underwent decision-making regarding Left Ventricular Device Assist (LVAD) treatment. Data comes from a larger study of patient and caregiver perspectives to inform the development of a decision aid to improve decision making about LVAD therapy. The data included in-depth interviews comprised of *outcome* narratives, or those addressing the psychological or physical outcomes associated with a decision, and *experiential* (“testimonial”) narratives that provide information about what it is like to have a given disease or treatment [12].

## 2. Experimental Section

### 2.1. Methods

#### 2.1.1. Sampling and Recruitment

We utilized theoretical sampling, which involves purposeful sampling in order to reflect representativeness of the LVAD population, while also obtaining a wide distribution of cases and experiences. Participants were recruited through patient rosters distributed once a week by the LVAD/program coordinator. All patients were asked before being referred whether they would be willing to talk with us, and if yes, their names were included on the roster. Participants were then approached before or during their appointments at the LVAD clinic or in the hospital without interfering with clinic flow and consent was obtained. The study was approved by the Institutional Review Boards of Baylor College of Medicine and The Houston Methodist Research Institute. Subjects were compensated for their interview time with a $25 gift card.

Participants included LVAD patients who had made the decision themselves about LVAD implantation, including those who accepted (*n* = 15) versus declined (*n* = 15) LVAD implant, LVAD candidates who had received education about the LVAD and were in the process of making a decision (*n* = 15), and caregivers (family or significant others) of LVAD patients (*n* = 15). Our sample included subjects across a range of age, ethnicity, socio-economic status, sex, amount of time post-LVAD, and sickness (Table 1). We employed no exclusion criteria.

Candidates were defined as NYHA Class III and IV patients, 30–80 years old and alert with decision-making capacity (as determined by the Aid to Capacity Evaluation (ACE)), with an acceptable surgical risk/benefit ratio for LVAD implantation (meaning that the likelihood of achieving the benefits of device therapy such as quality of life and projected survival improvement outweighed the projected risks such as early post-operative risk of dying, multi-organ failure, and failure to thrive on LVAD therapy as determined by a multi-disciplinary medical review board) and with good psychosocial support, coping mechanisms, and financial resources, as determined by administration of the Stanford Integrated Psychosocial Assessment for Transplantation (SIPAT) from the transplant social worker. “Decliners” were defined as individuals: (1) who described themselves as declining an LVAD; (2) whose refusal extended for a period of at least one month after the patient selection committee first approved LVAD placement; and (3) who met with consultants (e.g., bioethicists and psychologists who are members of the heart failure team) for the purpose of verifying that they were declining the therapy and to address any modifiable barriers impacting their refusal.

#### 2.1.2. Data Collection and Analysis

Structured interview guides were developed from researchers’ prior knowledge of domains and areas of interest, literature review, and expert opinion (see Blumenthal-Barby 2015 for full details) [13]. Domains were reviewed by clinical experts and included perceptions of options, outcomes, and probabilities; values in decision-making; degree of decision-making difficulty and factors contributing to difficulty; usual and preferred decision making roles; and decisional barriers and facilitators.

Interviews were audio-recorded, transcribed verbatim by an independent transcription agency and checked for accuracy. All investigators read three initial transcripts to provide insight about areas for potential exploration. Interviews were analyzed using ATLAS.ti [14] to systematically code, analyze, document, and interpret data, using thematic content analysis to deductively and inductively develop a collaborative codebook and find connections between themes by identifying salient quotations [15,16]. Codes were organized based on the five domains reported above in the interview guide [17]. Code assignments were made independently by team members, each generating new emergent codes as needed and later compared and discussed until consensus in coding styles was reached. Digital audio-recordings were deleted after transcription.

Frequencies and proportions for reporting themes were calculated using a denominator of 60 for pre-implant and 30 for post-implant themes, based on the assumption that the full sample had access to pre-implant experiences, while only patients (15) and caregivers of patients (15) had access to post-implant experiences, while decliners did not, and neither did candidates at the time of our interviewing.

## 3. Results

A majority (63%) of participants who accepted LVAD treatment shared “restitution” narratives that conveyed an overall shift from pre-implant physical suffering to post-implant experiences of healing (Table 2 and Table 3). Before deciding whether or not to receive an LVAD, 100% of the entire sample (including decliners and candidates who may not have gone on to receive an LVAD) experienced at least one form of suffering, including physical suffering, frequent hospitalization, limited mobility, low quality of life and negative psychosocial and emotional experiences. A majority (63%) of individuals who received an LVAD, including patients described by their caregivers, reportedly experience healing post-LVAD, suggesting overall subjective improvements resulting from LVAD therapy. Reported facilitators of healing included positivity (67%), acceptance and adaptation (77%), and social support (70%). While 100% of those who experienced healing always reported one or more of these facilitators, 95% of those who did not experience healing also reported at least one facilitator, suggesting these facilitators of healing are not determinative. Further, while narratives generally followed this restitution arc, a majority (86%) of participants experienced new challenges in the course of physical healing and adapting to new lifestyle changes and limitations. Subjective experiences related to these arcs of suffering and healing are elaborated below.

### 3.1. Suffering Pre-Implant

#### 3.1.1. Physical Suffering and Frequent Hospitalization

Overall, 65% of participants who described their pre-implant experiences reported physical suffering, with 63% mentioning frequent hospitalization. Patients describe this physical suffering with heart disease in terms of fatigue, dizziness, difficulties breathing, swelling of the limbs and body due to water retention, and the burdens of frequent hospitalization. One patient (15:31) commented, “With heart failure, you feel like hell […] the fatigue is incredible. It literally hurt. It was everything I could do to withstand the pain of just being fatigued and exhausted.” Another patient (30:11) shared, “It was very hard to keep my eyes open.”

In addition to fatigue, other patients reported having experienced difficulties in breathing. One patient (14:26) stated, “My health was steady going downhill. I was having these bouts with shortness of breath,” while another (63:6) similarly said, “My short walks to the restroom [were] becoming breathless. [My caretaker] and I both agreed that this is no quality of life. If they can’t fix this, this is it we will prepare to go home–*the* home,” referring to hospice.

Patients also described problems associated with water retention, primarily swelling. One patient (20:8) said “…my legs and feet swole up like elephant trunks,” while another (65:2) shared “[My legs] were just… like clumps. They were huge. I couldn’t walk. It was actually so bad that fluid was oozing out of my legs.” One patient 63:3 poignantly described, “I was so full of fluid, I was literally drowning in my own body.”

These symptoms frequently led patients and caregivers to the hospital, where they would be treated and discharged, only to be readmitted weeks or days later. One patient (77:2) commented, “I would go in the hospital, like on a Saturday, get out on a Friday, and then on Monday I’m right back in because I just kept being sicker and sicker.” Another (45:8) described, ““I was going into congestive heart failure every other week. I was always rushing to the hospital. One year I was in the hospital for a straight 7 months.”

Some patients were returning for the same procedures multiple times, as one decliner of LVAD (42:04) described, “I started having breathing problems, and I went to the doctor who drew off fluid, and once the fluid was off, a couple days later, I had to come back in and do it again.” In addition to the pain and suffering of heart disease itself, patients also reported the discomfort of being repeatedly “poked and prodded” in the hospital. One patient (75:9) shared, “I had fluid in me, two heart caths and two ventral lines and was stuck 42 times before they realized that my IV wasn’t going to hold enough central line.” Another (81:5) said, “They tried different things, – defibrillators, this, that and the other, [then] sent me to rehab, back and forth, back and forth.”

#### 3.1.2. Limited Mobility and Quality of Life

Patients discussed how the physical symptoms described above often resulted in limited mobility (52%) and negatively impacted their quality of life (43%). One patient (15:32) described, “With congestive heart failure, there is no movement. You are a zombie sitting there. It is so bad that it’s easier to keep your eyes closed than hold your lids open.” A caregiver (24:26-34) says of her husband, “He was dying. He was tiny and losing weight and could not walk. He was falling a lot at the house, because he couldn’t move his feet.” Patients described how declines in mobility affected their quality of life, especially spending time with family. A candidate (11:12) explained how his immobility inhibited his ability to enjoy family outings, saying that he was always “…having to worry about shortness of breath, fatigue, or [that] I′ve got to stay in the hotel room because I can’t walk 50 feet and be winded. Then that ruins the vacation. It’s not a family thing anymore.” Another patient (60:6) similarly commented, “I want to enjoy my family and friends and just continue on with my life, but as I’ve gotten weaker, I’ve been able to do less and less.”

#### 3.1.3. Negative Psychosocial and Emotional Experiences

The physical limitations of advanced heart failure often carry over into negative impacts on mental health and well-being. Negative emotional states range from depression or disappointment associated with changes in roles to worry and anxiety (“scared” and even “terrified”) about worsening of symptoms and uncertainty about treatment options and outcomes. One patient (22:17) said, “You’re depressed because you can’t do anything. You feel ill all the time.” These negative thoughts can also be compounded by cognitive deficits that are experienced as frightening and debilitating. One candidate (65:1) explained:
I will have conversations [that] I forget. I just forget what I’m talking about. The past two weeks, I was out of it pretty much. That’s why it was freaking me out. Everybody had to talk for me because I was just blank.

Another patient described his (27:31) pre-implant experience:
I was not getting enough oxygen to my brain. I was actually stupid, and I was refusing to go back to the hospital. I did not understand what my situation was. It was so bad that I could not get up from the couch and make it to the bathroom. I wasn’t getting oxygen anywhere in my body.

Patients reported a common reluctance to return to the hospital, for fear of receiving more bad news. One decliner (44:32) said: “Every time I came to the hospital, [the doctors] were like, ‘It’s getting worse. You’re going to have to stay in here for a while,’ and I’m like, oh man, I got to go to work tomorrow. What am I going to do?”

The common inability to maintain work roles and responsibilities were also experienced negatively by others. As one patient (22:17) shared:
I’ve always been the man of the house, bringing home the money, taking care of the kids. That’s one thing you might not be able to do. In my case I can’t, and it depresses you a little bit. Because your kids are growing, you can’t help out financially or anything because you can’t work. And your disability [pay] is not going to be as much as you’re used to making, so it depresses you a little bit.

Another candidate (58:80) described negative emotions in relation to his inability to stay involved in family activities:
[Usually] I play with my two little girls, and I’ll help my wife fix dinner or my kids with homework—[but] at one point in time I just—I was depressed, and I just gave up. I don’t feel good at times.”

### 3.2. Healing Post-LVAD

#### 3.2.1. Reclaimed Mobility, Independence and Quality of Life

A majority (63%) of our participants’ narratives shift from stories of pre-implant disability, depression and anxiety to post-implant healing experiences, including substantial gains in mobility and quality of life after receiving an LVAD. One patient (13:20) shared, “Since they put the LVAD, I don′t have that pain no more. And I walk, I do things.” Others explained, “I was getting to where I couldn’t walk nowhere. Before I got there, I was just wore out. I was in bad shape. But I’m right now,” (17:30) and “Before surgery, I could not walk as far as I can now without getting short-winded. I really don’t get short-winded now. In that sense, it’s better.” (21:17) Another (77:6) attested, “All in all, the LVAD, it helped me, because it’s helping me to breathe. I get around a little better, you know, I’m feeling all right.”

One patient (74:12) said of the transition, “It will make your quality of life better. I don’t run out of breath anymore. It’s a matter of having regular stamina. At first, I couldn’t walk a block.”

Patients recounted a sense of reclamation over their lives. Getting the LVAD helped a number of participants return to work, get back to their hobbies and engage in routine activities like cleaning the house. One patient (21:18) said that after getting the LVAD, “I work full time, and I can go out for a walk, or I can go to the rodeo and walk that whole parking to get to the door. Before it would take me a half a day.” Another commented that post-implant, “I clean my house, cook for myself, a lot of things–personal things, and I’m really proud of it” (13:25). A patient who had been frequently rehospitalized likewise said, “I’m not in the hospital as much, and doing better. It’s brought me back on my degree of quality of life.”

Many patients also reported positive psychosocial changes, such as increased independence after receiving an LVAD. One patient (79:13) proudly shared that “I can drive by myself a bit [now],” while his caregiver added, “The look on his face, to be able to walk into the church, to walk on his own. He goes, I got my job back [as an usher]! One caregiver (34:22) described her husband as able “to do just about anything. I just let him do what he wants to do. If he needs help or assistance… I’m right there behind him. He does what he wants to.” Another (24:20) shared of her husband, “[The LVAD] really changed our life. He’s gone from not being able to do anything to back to where he was, going fishing, mowing the lawn, washing the cars. He’s staying by himself now. He does everything for himself.”

#### 3.2.2. Adapting to New Challenges

These positive gains in independence and mobility are embedded within a larger process of healing that for many participants (87%) came with substantive challenges. Some participants pointed out that pain and suffering do not simply disappear right away and highlighted difficulties dealing with post-implant pain within the first few months after surgery. One patient (54:23) described, “The first 30–45 days are going to be hell. Because the operation, the ICU, recovery, the big displace of home, family problems. Your life change is going to start right off the bat.” Another patient (57:12) commented, “I was about forty days just flat on the bed. I couldn’t get up. It was a very difficult recovery period after the operation.” One caregiver said of her husband after LVAD surgery (32:16), “We had to dress him, clean him, diapers, the whole thing. Bathe him, he couldn’t get up, he couldn’t raise his arms up.” One patient (20:11) vividly describes her post-operative experience, saying “I woke up and I felt the incisions that they made. I had a very dry, uncontrollable cough, and every time I coughed, it felt like they were pulling apart—that hurt real bad.” While many patients transition from this period of recovery with few impediments, others (*n* = 16, 27%) experience continued complications, such as gastrointestinal bleeding or infection, that lead them back into the hospital. One caregiver (41:13) of a post-implant patient said, “Ten days after leaving the hospital, we’re back. He’s bleeding again… It gets me that we have to keep coming back. We’ve been back twice for the bleed. That’s what is frustrating.”

Other new challenges include carrying and managing the batteries that supply power to the LVAD, the inability to shower, bathe or immerse oneself in water due to high chance of driveline area infection, the inadvisability of driving for some due to lower maneuverability and potential damage from airbags. The challenge of accepting these new limitations as permanent are evidenced by a comment from one patient (19:27) who said, “You don’t have an option of ‘I’ll just wear that today and not tomorrow,’ or ‘I can wear it for 20 min today and not tomorrow.’ You have to be really diligent.” Some participants reported difficulties sleeping due to discomfort related to sleeping with the battery. “I can’t sleep on my side,” said one patient (16:7), “because if I move, I have to move [the battery] to sleep.” Other patients complained that the battery” becomes heavier and heavier” every day 57:11), and that “if you’re at home, you have to stay hooked up to the wall.” (82:4). Another patient (16:7) commented, “I have to carry the control and the batteries. I can’t go to the pool. I can take a bath, but it’s difficult. I have to prepare everything,” referring to the extensive measures necessary to keep the driveline dry. Other respondents similarly mentioned the limitations imposed by the inability to full immerse in water. One patient (62:5) said, “It would be nice to take a normal bath without a long ritual when you get ready to shower. It takes so much longer now.”

### 3.3. Facilitators of Healing and Adaptation

While these new challenges are experienced as formidable for LVAD patients and caregivers alike, respondents pointed out certain factors that facilitate healing and adaptation, including: (1) positivity (67%); (2) acceptance and adaptation (77%); and (3) social support (70%).

Maintaining positivity and being determined to continue living a high quality life were viewed as catalysts for healing. Advice from one patient (24:35) was “Don’t be afraid. You’re going to be okay. It’s not as hard as it sounds. And it doesn’t take up as much of your life as you think it does. Once you get into your routine of life, there’s really not much to it, you know?” Maintaining positivity was related to seeing a “silver lining,” as described by one caregiver (49:49): “It’s not going to be perfect, but we’re going to be able to go and travel and sing at our church on Sunday and not have to carry her oxygen around with her.” Another patient (20:40) said healing post-LVAD is related to finding a purpose: “I’m determined to go ahead and continue with it because it’s keeping me alive and I’m loving every minute that I’m spending with my kids.” Another (45:18) voiced his determination by saying: “I had to make up my mind. ‘You’ve had this surgery. Now you’ve got to get back. You’ve got to fight.’”

Participants described a need to find acceptance of new challenges, including new daily routines and even new perceptions of oneself. One patient (20:10) admitted, “I was upset about the whole process. When I came out of surgery, I didn’t like the idea of a machine attached to my heart.” Another (31:14) said, “I didn’t like it [at first] either. I mean, who wants a six-pound bag hanging off your shoulder all the time? But when you realize what it does, and it’s keeping you alive, then it’s a no-brainer, you know? It’s just something you have to get used to.” Another (45:17) shared, “It’s not easy, but you get adjusted to things because this is your life. This is what keeps you going.” One participant (15:29) equated this adjustment period to facing another of life’s stages:
It’s no different than when you are a baby and all of a sudden you’ve got to start brushing your teeth. It’s just a new process, and I have got it down now. I can shower, shave, change my bandage, change the batteries in 45 min flat, and that’s pretty quick. You just get after it.

Another respondent provided an example of how personalized adaptations can help individuals to acclimate to wearing the LVAD. Having customized his own battery-support backpack, he explained, “I had to really learn how to make the most of what I have here, and I think the backpack was a big deal for me, because now I can have both hands free.”

Support from caregivers and other family or community members was a common key factor in fostering patients’ positivity and strength. A caregiver pointed out that sometimes “You have to force [your loved one] to make the LVAD part of their life and do that things that they can’t do on their own.” Similarly, another said, “My kids motivated me to stick with it. There have been times I didn’t feel like coming to the clinic or to doctor’s appointments, and they were the ones who motivated me to do it.” Another caregiver (24:31) explained how demonstrating strength and determination helped to elicit the same in her loved one:
I think having that positive realm about him is helpful for him. If you have people that are scared all the time, then that’s how you’re going to be. But that’s not how our family is. We take care of each other. We’re always there for each other and so I think that helped him know that everything was going to be okay.

Caregivers can also help to manage a loved one’s emotional states by recognizing and encouraging productive versus destructive emotions. As one caregiver (62:6) pointed out, “Sometimes he will go into pity mode, and that doesn’t do anything.” Caregivers, too, pointed out that the recovery and adaptation process is challenging, but soon becomes “everyday habit” (41:15). One caregiver (24:33) described, “When we first got home, it took a while to do everything. But now we’re just like (snapping sounds), we can get up, we can go, we can get it connected.” Some patient and caregivers find additional support through social media. One respondent (63:8) said, “Social media has kept me strong. My Facebook friends and family on there are constantly bombarding me with good things to talk about and prayers. That keeps me strong.” Over time, these forms of support help patients to “stand up, and face it,” (37:32), referring to the challenges of healing and recovery. As one caregiver said of her loved one, “[Eventually] everything just kind of got back in place. Got out of bed, out of the wheelchair, off the walker, and then started back just like he used to.”

## 4. Discussion

The narratives described here follow a shared arc, beginning from a bleak period of disability, including disassociation with a prior sense of self and debilitating fear and anxiety about the future, into one of healing and reclamation. While others have argued that this “arc” and other narrative frameworks may be imposed upon rather than emerging organically from the data [18], we believe that certain elements intrinsic to the data give us reason to characterize the LVAD healing experience as commonly arc-like. Specifically, they are progressive rather than regressive [19], with a majority of LVAD experiences involving improvement and healing rather than deterioration, continued suffering, or decline over time. Participants reported lower quality of life, greater physical suffering, more frequent hospitalization and more negative psychosocial experiences pre- versus post-implant. While few other studies qualitatively examine subjective changes resulting from LVAD treatment, Otternberg et al. [20] and Chapman et al. [21] discovered similar subjective improvements in functional status and symptom burden, though these studies are somewhat dated. These findings support an interpretation of an arc-like trajectory to patients’ experiences, with LVAD implantation both the objective and subjective turning point in reclaiming one’s former activities and sense of self. These patients’ experiences fit the restitution narrative, first described by Frank [2], that describes an individual’s steps to return to a previous state of health and to restore one’s damaged sense of self. Disease is viewed as an enemy to battle against, and from which to reclaim the self, to do things “just as [one] used to do,” in one caregiver’s words. This element of reclamation may be especially an important feature in Western settings where individuality and self-sufficiency are highly valued [22]. The narratives reported here reveal a common encounter with states of being experienced as unbearable (e.g., “literally drowning in my own body”) in which individuals feel physically and emotionally disassociated with their former selves and incapable of performing usual roles and responsibilities. Often times, even one’s body becomes uncooperative and unrecognizable as one’s own (e.g., with legs and feet that “swole up like elephant trunks,” and another patient comparing her legs to huge, inanimate “clumps.”)

### 4.1. Structuring Illness Experiences through Narrative

These are not simply descriptions of what it is like to have heart failure pre- and post-LVAD. The “narratives about illness” described here may also constitute “illness as narrative” more broadly, as described by Hyden [23], in which experience is seen through the lens of an illness, which helps to structure the problems and resolutions that a patient experiences subjectively. In the case of LVAD, the loss of functioning and the associated departure with the former self is inextricably couched within the particular experience of trying to overcome symptoms of heart failure through receiving an LVAD. This entails facing fears about surgery and post-operative pain and suffering, fears about losing one’s independence and burdening caregivers, and coming to terms with the notion of having a foreign object (“as big as a coke can,” said one of our respondents) in the body. Moreover, as our participants pointed out, the acceptance of an LVAD is not just the acceptance of a treatment or a particular medical device, but an acceptance of new lifestyle changes, new responsibilities for the self and for caregivers, and even a new identity, or what some of our respondents called a “new normal.” In this sense, the experience of receiving an LVAD is more than simply receiving a treatment that “cures” someone of their illness; instead, it forms part of a larger, more dynamic process of healing and adaptation in which an individual continuously negotiates the expression of one’s ideal, former self within the practical limitations of one’s new state of being. Rather than telling a one-dimensional story about overcoming heart failure, the individuals we interviewed recounted a shared experience of recovering former selves and lifestyles, along with former meaning and purpose, where receiving an LVAD constituted the narrative “pivot point.” Our respondents described this recovery as facilitated by certain psychosocial factors, including maintaining a positive and adaptive approach to healing, as well as receiving assistance with self-care and social support from family or friends. These facilitators, particularly the importance of reclaiming aspects of one’s identity through self-reflection and social support, have likewise been noted by Slocum et al. [24].

### 4.2. Function and Use of Narratives in the Clinical Setting

These narratives also move beyond description in that they *do* something for patients. Stories “reopen a space,” writes Morris (2008) “where physician and patient may participate in an engagement that transforms the humdrum day-in/day-out diagnostic encounter into a source of truly sustaining, restorative satisfactions” [25]. For a patient, recounting a narrative can be cathartic, helping to unearth personal meaning and significance from an illness experience [10,26,27,28]. As Blaxter (2010) pointed out, in this view of narrative, “Stories do not just describe the experience; they are repair work, creating a new self” [29] (p. 65).

The use of narratives in clinical settings can also provide useful guides for decision making [5,7], helping patients better envision treatment processes and outcomes by framing them in the context of a narrative structure, plot and resolution [30]. For LVAD patients, what is lost in the course of suffering from heart failure (i.e., the former self and one’s normative relations to others) can be seen as gradually regained post-implant. Illnesses like heart failure are rarely conceptualized subjectively by individuals as a “thing,” but instead as an integral part of a person’s being and experience [2]. Thus, relying exclusively on biomedical information may inhibit patients from “building a picture” of the various elements perceived as important for their decision making (e.g., experience of pain or discomfort, identity continuity). Narratives, on the other hand, draw attention to other aspects of the social and clinical environment, holistically combining clinician feedback with a patient’s own intuitions and awareness of others’ experiences and outcomes [31,32].

It is important to note that narratives can also be biasing, and patients and caregivers should be aware that narratives are non-generalizable. Research suggests that targeted experience narratives (targeting the direction of forecasting bias) can help patients to more accurately forecast health outcomes (e.g., predicted discomfort) than representative narratives, or those that describe the full range of possible experiences [5]. While this may be true for some conditions, in the case of LVAD specifically, evidence has shown that a decision aid for LVAD therapy which included narratives recounting both positive and negative experiences was associated with greater patient knowledge about what to expect, how to manage daily life with an LVAD, and greater overall life satisfaction post-implant [33]. When used in the context of decision-making, clinicians are advised to strategize the role of narratives in ways that maximize patient decisional needs and avoid undue bias. Further, while we have taken every precaution to avoid imposing an arc structure (e.g., from suffering to healing) upon the data (e.g., by inductively and collaboratively identifying themes, by comparing theme frequencies pre- and post-implant), we acknowledge the possibility of bias in our own understanding of the data, as some critics of narrative have put forth [34], and encourage clinicians to alert LVAD candidates and caregivers that alternative narratives exist. The very notion that someone’s experience can and should fit into a narrative format at all (rather than, e.g., metaphors [35], or non-linguistic modes of expression [36,37]) has been critiqued as an imposition of Western middle-class, liberal and neo-liberal modes of being [38,39]. As Woods [40] and Strawson [41] pointed out, not everyone is “naturally narrative.” We, too, acknowledge a possibility that the events, emotions and understandings we present here may not be experienced as narratives (“arc”-like or otherwise) by our respondents, even if we as researchers and clinicians understand and convey them as such. We do, however, believe in their clinical utility in light of evidence that patients and caregivers in the LVAD community express appreciation for narratives in the course of their decision-making about and recovery from LVAD implantation [33]. Further, not all patients must understand their experiences as narratives for them to glean useful information from other patients’ experiences. Nevertheless, encouraging patients and caregivers to exchange experiences must be accompanied by an awareness that narrative is a powerful tool to be employed with care and self-reflection.

From the clinician’s point of view, narratives may help to better understand the logic behind discrepancies in decision-making between patients and clinicians. One example is that, while some patients (and indeed clinicians) may view the LVAD as a “savior,” others may object to the device itself as a “dehumanizing” foreign object that interferes with a sense of bodily integrity [42]. Understanding the varieties of patient perceptions and experiences through the stories patients and caregivers tell can increase empathy, reflection and trust between patients and their care providers [4], and can help clinicians to foster better patient-practitioner communication and understanding [43]. Studies show that better communication in the form of quality shared decision-making [44], patient-centered care [45], and adequate conveyance of clinical information [46] can improve patient satisfaction with care, disease-specific knowledge, more accurate self-reported health status, greater self-efficacy and adherence to self-management behaviors, emotional health, recovery from discomfort, and reduction in referrals for diagnostic testing [47,48,49,50,51]. Poor communication, on the other hand, is linked with patient distrust, disenrollment in health plans and clinics, malpractice litigation, less health-seeking behaviors and use of preventive services, and low adherence to medical advice among patients [52].

While clinicians may not have time to elicit lengthy narratives from every patient, an awareness of common trajectories pre- and post-implant may help clinicians to better understand shared experiences and subjective needs of their patient population. Clinicians can offer further support by referring their patients to other patients whose narratives may offer recognition, affirmation and understanding of their experiences, including doubts and fears about treatment. Likewise, they can make published narratives such as those offered here available to their patients. For more patient and caregiver narratives specific to LVAD, see www.lvaddecisionaid.com.

## 5. Conclusions

Sharing illness narratives, including both positive and negative aspects or treatment and recovery, can help patients and clinicians to openly discuss expectations and reservations about treatment options. Narratives can foster understanding between patient and clinicians, with potential benefits for maintaining positive health behaviors consistent with clinician recommendations and patient needs. Patients can also benefit from hearing other patient narratives in order to help manage expectations, forecast outcomes and derive personal meaning from their own experiences.

## Figures and Tables

**Table 1 jcm-08-00660-t001:** Characteristics of sample: LVAD patients, candidates, and caregivers.

Characteristic	LVAD Candidates (*n* = 15)	LVAD Patients * (*n* = 15)	LVAD Caregivers ^y^ (*n* = 15)	LVAD Decliners (*n* = 15)	Overall (*n* = 45)
Age, mean	54 years (range 35–74)	60 years (range 33–74)	59 years (range 36–74)	60 years (range 45–82)	58 years (range 33–74)
Male	13 (87%)	11 (73%)	5 (33%)	10 (67%)	29 (64%)
Female	2 (13%)	4 (27%)	10 (67%)	5 (33%)	16 (36%)
Ethnicity					
White	6 (40%)	7 (47%)	9 (60%)	7 (47%)	22 (49%)
Black	3 (20%)	6 (40%)	5 (33%)	7 (47%)	14 (31%)
Hispanic	6 (40%)	2 (13%)	1 (7%)	1 (7%)	9 (20%)
Reason for LVAD					
Bridge to Transplant	NA	2 (13%)	1 (7%) ^z^	5 (33%)	3 (10%) ^x^
Destination Therapy	NA	13 (87%)	14 (93%) ^z^	10 (57%)	27 (90%) ^x^
Hospital Status					
Inpatient	14 (93%)	6 (40%)	4 (27%) ^z^	7 (47%)	24 (53%)
Outpatient	1 (7%)	9 (60%)	11 (73%) ^z^	8 (53%)	21 (47%)
Length of time with LVAD mean	Not implanted	539 days (range 16–1894)	634 days ^z^ (range 50–1845)	Not implanted	586 days ^x^ (range 16–1894)
Monthly Household Income, mean	$3171 (range $423–10,833)	$2383 (range $528–7916)	$2600 ^z^ (range $961–6250)	$2645 (range $739–8333)	$2730 (range $423–10,833)

LVAD, left ventricular assist device; NA, not applicable. * All 15 patients received a HeartMate II continuous flow LVAD; ^y^ The term “Caregiver” corresponds to the patient’s spouse or family member, which primarily supports the patient; ^z^ Designates characteristic of LVAD patients supported by caregivers; ^x^
*n* = 30.

**Table 2 jcm-08-00660-t002:** Variables of suffering and healing pre-implant.

	Patients	Caregivers	Candidates	Decliners	
	*n* = 15	*n* = 15	*n* = 15	*n* = 15	TOTAL = 60
Spontaneous discussion of suffering	10	3	12	12	37
	67%	20%	80%	80%	62%
Physical Suffering	14	7	6	12	39
	93%	47%	40%	80%	65%
Frequent Hospitalization	9	8	10	11	38
	60%	53%	67%	73%	63%
Limited Mobility	12	3	7	9	31
	80%	20%	47%	60%	52%
Low Quality of Life	4	3	11	8	26
	27%	20%	73%	53%	43%
Negative Psychosocial and Emotional Experiences	6	1	11	9	27
	40%	7%	73%	60%	45%

**Table 3 jcm-08-00660-t003:** Variables of suffering and healing post-implant.

	Patients	Caregivers	
	*n* = 15	*n* = 15	TOTAL = 30 *
Experienced Healing (Improved Mobility, Independence and Good Quality of Life)	12	7	19
	80%	47%	63%
New Challenges	11	15	26
	73%	100%	87%
Facilitators of Healing:			
Positivity	8	12	20
	53%	80%	67%
Acceptance and Adaptation	9	14	23
	60%	93%	77%
Social Support	10	11	21
	67%	73%	70%
Experience Healing and at Least 1 Facilitator	12	7	19
	100%	100%	100%
Did NOT experience healing but Reported at Least 1 Facilitator	12	6	18
	100%	86%	95%

* The post-implant denominator does not include candidates or decliners.

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
