# Peer review of "Suffering and Healing in the Context of LVAD Treatment"

_jcm, 2019, doi:10.3390/jcm8050660_

Reviewer 1 Report

This is a well presented clear case for the use of narrative techniques in clinical medicine. The examples are well described, and the research was thorough and clearly yielded rich results. The arguments for using so-called 'narrative' approaches are compelling, and I find the paper to be significant and to have a high overall merit because of the importance of this topic.  The only drawback I found in the paper is that 'narrative' is used in a rather naïve way, and there is little reflection on the issues raised by narrative - for example, what counts as a narrative rather than as a rendering of qualitative research. In reading the descriptions of the interviews, I was left wondering why they were particularly narrative in form, what makes these narrative rather than descriptions of episodes? They are described as all having the a 'shared arc', that of the 'restitution narrative' described by Frank. I was not convinced that this is not a narrative shape that is imposed in giving form to the data from the interviews rather than something that necessarily is any particular patient's narrative. I find this 'shared arc' surprising given the diversity of participants in the study (lines 75-76) and I wonder if this might not come from some marshaling of the evidence into a particular narrative shape (Note Woods's criticism of Frank, for example).  I also note that the experiences of the decliners, who presumably do not have 'restitution narratives' are not discussed again -- why not? There is only one reference to the potential bias of narrative, and this is not addressed. Giving the results of interviews a narrative form is not neutral, it quite strongly shapes how the interviews are reported, and if used in the ways that the author(s) suggest, can also shape the experiences of patients rather than merely reporting them.  This could amount to an ethical harm. I think that the paper would be improved by showing at least awareness of some of the issues with narrative. I would suggest a reading of Angela Woods, 2011, The Limits of Narrative: provocations for the medical humanities. doi:10.1136/medhum-2011-010045, which also has several more references that would be useful.

Author Response

We appreciate the opportunity to receive valuable feedback from our reviewers who took the time to offer constructive suggestions for revision. We have carefully considered and integrated their suggestions into our revised manuscript. Our responses to their specific concerns are detailed below.

Reviewer 1:

Reviewer 1 brings up three main important points (bolded), that we discuss in turn, following his/her comments below.

This is a well presented clear case for the use of narrative techniques in clinical medicine. The examples are well described, and the research was thorough and clearly yielded rich results. The arguments for using so-called 'narrative' approaches are compelling, and I find the paper to be significant and to have a high overall merit because of the importance of this topic.  The only drawback I found in the paper is that 'narrative' is used in a rather naïve way, and there is little reflection on the issues raised by narrative - for example, what counts as a narrative rather than as a rendering of qualitative research. In reading the descriptions of the interviews, I was left wondering why they were particularly narrative in form, what makes these narrative rather than descriptions of episodes? They are described as all having the a 'shared arc', that of the 'restitution narrative' described by Frank. I was not convinced that this is not a narrative shape that is imposed in giving form to the data from the interviews rather than something that necessarily is any particular patient's narrative. I find this 'shared arc' surprising given the diversity of participants in the study (lines 75-76) and I wonder if this might not come from some marshaling of the evidence into a particular narrative shape (Note Woods's criticism of Frank, for example).  I also note that the experiences of the decliners, who presumably do not have 'restitution narratives' are not discussed again -- why not? There is only one reference to the potential bias of narrative, and this is not addressed. Giving the results of interviews a narrative form is not neutral, it quite strongly shapes how the interviews are reported, and if used in the ways that the author(s) suggest, can also shape the experiences of patients rather than merely reporting them.  This could amount to an ethical harm. I think that the paper would be improved by showing at least awareness of some of the issues with narrative. I would suggest a reading of Angela Woods, 2011, The Limits of Narrative: provocations for the medical humanities. doi:10.1136/medhum-2011-010045, which also has several more references that would be useful. 

This reviewer’s first comment is that our use of the term narrative neglects an acknowledgement of certain criticisms of narrative, especially those discussed in Wood (2011). This reviewer offered the astute observation (which we agree with) that some narrative structures are imposed upon rather than discovered in the data. We would like to clarify that our interpretation of the narrative structure stems from an inductive interrogation of the data, using a codebook that was arrived at inductively and collaboratively, meaning we reached a consensus – systematically – about what we separately saw emerging from the data. We did this in an earnest attempt to reduce the potential for researcher bias, and believe that our interpretations of the overall patterns across participants represent a shared experience of moving from suffering to an experience of healing. We do note that even healing experiences often entail some forms of suffering (we characterize them as “challenges”); however, we found many more spontaneous discussions of unresolved suffering in descriptions of pre-implant experiences, versus more spontaneous recounts of healing post-implant.

Nevertheless, we do think it is appropriate to draw attention to the criticisms that have been laid out regarding the possible imposition of order on unstructured narratives, including some cited in Wood’s (2011) article and carried over into ours (see lines 329 and 404). We have included some explanation to this effect in the first paragraph of the discussion (line 329) and have offered our view of certain elements intrinsic to the data that give us reason to believe that our observed arc structure is not an imposition of researcher bias.

Because we believe that the arc format is not something we have imposed on the data, we have not discussed the potential for ethical harm to participants, specifically, but we do acknowledge in our revision that there is a possibility that alternative narratives exist, and urge clinicians to make this clear to LVAD candidates and caregivers (see line 400). We also draw attention to the proposition that for some individuals, narratives are not perceived as useful vessels for understanding (Sartwell 2000).” (line 404)

                  We hope these revisions satisfy the reviewer’s helpful suggestion to “show at least awareness of some of the issues with narrative.”

Reviewer 2 Report

The paper presents narratives of suffering and healing in patients pre- and post-LVAD implantation.

This work was very informative and interesting to read. It's commendable to study and report the patients viewpoint, and this can motivate a lot of the engineering and design work related to improving LVADs (specifically reducing the battery weight or moving to transcutaneous charging of implantable batteries). I commend the authors on presenting this patient-centric work to the scientific community.

Although the narratives are extremely interesting, I feel that they could be presented in a more organized fashion.

Would it be possible to cluster narratives from patients, and organize them, for example by subtheme in a table?

Could the authors quantify the data - for example out of the n=15 from each group, how many expressed feelings of depression/dismay/frustration? How many mentioned the words dying/inconvenience/recovery. (these are all examples and I'm sure that your algorithms could do a better job at identifiying appropriate metrics)

If these could be quantified and plotted in bar charts and compared this might be helpful for the reader to get an overview of the feelings that patients/caregivers had towards this.

It might also be interesting to compare the patient vs caregiver take on a situation - did they mention the same issues or benefits or did they prioritize different things?

I think the authors should try to integrate the narratives, process them and make solid summaries and conclusions based on the data collected, instead of a myriad of patient quotes.

The conclusionin the abstract is very similar to in the manuscript.

For LVAD patients, other patients’ illness narratives

23  can help to give meaning to their own illness and treatment experiences and to more accurately

24  forecast treatment impacts on lifestyle and identity. For clinicians, patient narratives can enhance

25  patient-practitioner communication and understanding by highlighting perspectives and values

26  that structure patients’ clinical experiences.

Neither can be directly interpreted from the results presented. While I agree that it is a reasonable assumption , it should be substantiated. Can the authors use these narratives to inform patients/clinicians and collect feedback on that? Even a binary helpful/not helpful or insightful/not insightful from a group of each would substantiate the conclusion.

Author Response

We appreciate the opportunity to receive valuable feedback from our reviewers who took the time to offer constructive suggestions for revision. We have carefully considered and integrated their suggestions into our revised manuscript. Our responses to their specific concerns are detailed below.

Reviewer 2:

 Reviewer 2 offered three main suggestions for our paper, which we address in turn below.

1. Although the narratives are extremely interesting, I feel that they could be presented in a more organized fashion. 

Would it be possible to cluster narratives from patients, and organize them, for example by subtheme in a table?

Could the authors quantify the data - for example out of the n=15 from each group, how many expressed feelings of depression/dismay/frustration? How many mentioned the words dying/inconvenience/recovery. (these are all examples and I'm sure that your algorithms could do a better job at identifiying appropriate metrics)

If these could be quantified and plotted in bar charts and compared this might be helpful for the reader to get an overview of the feelings that patients/caregivers had towards this.

We have taken this reviewer’s suggestion to organize our presentation of themes and sub-themes (i.e. variables related to suffering and healing pre- and post-implant) into Tables 2 and 3. While we did not plot them in bar charts, the tables present the frequencies in straightforward fashion, allowing for comparison within and across groups. We hope this clarifies not only the content but also the frequency of themes across our respondent sample. We did so in response to this reviewer’s request to more clearly quantify the data in relation to the themes that emerged organically (i.e. inductively). We chose this inductive approach over, say, the more deductive approach of searching for a priori keywords, as the reviewer proposed (and which is both appropriate and productive in some other cases) in order to be more consistent with our chosen qualitative research methodologies which remain true to the inductive paradigm (see our citations 14 and 15).

2. It might also be interesting to compare the patient vs caregiver take on a situation - did they mention the same issues or benefits or did they prioritize different things?

While the purpose of this paper was not to compare patient vs. caregiver experiences, but rather to explore patient experiences before and after LVAD (directly through the patients’, candidates’ and decliners’ recounts, or indirectly through caregivers’ eyes), we do present caregiver perspectives on the same variables, allowing the reader to compare caregiver vs. patients’ perspectives. For a more in-depth view of caregiver experiences in relation to LVAD therapy, we kindly refer the reviewer to the sizable literature on the topic, including three recent articles below, including one (the first in the list) from our research team:

Bruce, Courtenay R., Charles G. Minard, L. A. Wilhelms, Mackenzie Abraham, Javier Amione-Guerra, M. S. S. W. Linda Pham, Sherry D. Grogan et al. "Caregivers of Patients With Left Ventricular Assist Devices." Circulation: Cardiovascular Quality and Outcomes 10, no. 1 (2017).

Magid, Molly, Jacqueline Jones, Larry A. Allen, Colleen K. McIlvennan, Katie Magid, Jocelyn A. Sterling, and Dan D. Matlock. "The Perceptions of Important Elements of Caregiving for an LVAD Patient: A qualitative meta-synthesis." The Journal of cardiovascular nursing 31, no. 3 (2016): 215.

Cicolini, Giancarlo, Francesca Cerratti, Carlo Della Pelle, and Valentina Simonetti. "The experience of family caregivers of patients with a left ventricular assist device: An integrative review." Progress in Transplantation 26, no. 2 (2016): 135-148.

3. I think the authors should try to integrate the narratives, process them and make solid summaries and conclusions based on the data collected, instead of a myriad of patient quotes.

This comment largely overlaps with the reviewer’s 2nd  suggestion to more systematically present our findings. We certainly wish to present our findings clearly, given our involved process of arriving at them. We employed a rigorous and systematic methodology to ensure that our summaries and conclusions are solidly based on the themes that emerged organically from our data. These themes were identified through an exacting process of qualitative analysis that included independent identification of themes across members of our research team using qualitative data analysis software (ATLAS.ti), iterative rounds of consensus about the existence and characterization of those themes in the data, systematic coding of all interviews in search of these and any other emergent themes that might arise, and finally, conscientious organization and presentation of themes into summaries, represented by the main and sub-headings of our Results section (e.g. Suffering Pre-Implant: Physical Suffering and Frequent Hospitalization, etc.). For clarity, these Results headings also clearly correspond to the variables of suffering and healing presented in our Tables 2 and 3. As such, we feel strongly that our results represent more than just “a myriad of patient quotes,” and hope that the reviewer agrees that we made an earnest effort to clarify our summaries and conclusions via the addition of these new tables, along with our discussion about how (and whether) to interpret meaning from these organized quotes (see Discussion beginning on line 329) .

Reviewer 3 Report

Kostick et al. examined the narratives of suffering and healing from 30 patients and their caregivers with advanced heart failure who engaged in decision-making regarding LVAD treatment. They suggested that for clinicians, patient narratives can enhance patient-practitioner communication and understanding by highlighting perspectives and values that structure patients’ clinical experiences.

The patient’s narratives are useful step forward towards a daily life of LVAD subjects. Nevertheless, I have a rather skeptical perspective on a fundamental level referring to the added value the paper provided from a scientific and a practical point of view. Authors need to do the major revision before the paper is considered for publication. Details are given below:

1)      Authors needs to check the number of patients.

L 71: Participants included LVAD patients who had made the decision themselves about LVAD implantation, including those who accepted (n=15) versus declined (n=15) LVAD implant…

L 114: A greater number of participants spontaneously discussed suffering pre-implant (n=40, 67%) versus post-implant (n=27, 45%), indicating overall subjective improvements as a result of LVAD therapy. 

2)      Table 1 is not attached.

3)      Authors needs to make a new Table to compare the narratives in patients with accepted versus declined LVAD implantation in terms of the 1) positivity, 2) determination and 3) support. In addition, authors need to make a quantitative, statistical analysis.

4)      Exclusion criteria should be added.

5)      Please compare this research with the other similar results available in the literature.

6)      Please add limitations of the study using separate limitations section.

Author Response

Reviewer 3: 

Reviewer 3 offered three main suggestions for our paper, which we address in turn below.

1) Authors needs to check the number of patients.

L 71: Participants included LVAD patients who had made the decision themselves about LVAD implantation, including those who accepted (n=15) versus declined (n=15) LVAD implant…

L 114: A greater number of participants spontaneously discussed suffering pre-implant (n=40, 67%) versus post-implant (n=27, 45%), indicating overall subjective improvements as a result of LVAD therapy.

We are grateful to this reviewer for drawing attention to some discrepancies in our reported frequencies and have made changes throughout the manuscript to reflect our recalculations. We have also clarified these frequencies and percentages in two new tables (Tables 2 and 3), and hope this resolves this reviewers’ concern. 

2)      Table 1 is not attached.

3)      Authors needs to make a new Table to compare the narratives in patients with accepted versus declined LVAD implantation in terms of the 1) positivity, 2) determination and 3) support. In addition, authors need to make a quantitative, statistical analysis. 

We have attached Table 1 (please excuse us for missing the attachment before), along with new Tables 2 and 3 which present our emergent themes and sub-themes (i.e. variables related to suffering and healing pre- and post-implant) in a way that we hope is clearer and more organized than before. We would like to draw the reviewer’s attention to the frequencies we have provided, which allow for comparison within and across groups. We did not conduct a statistical analysis of differences because the primary aim of our paper was to descriptively and qualitatively (not quantitatively) describe patients’ experiences of suffering and healing related to LVAD therapy. Our qualitative approach fills a gap in the literature on LVAD patient experiences, as the existing literature tends to focus on quantitative representations of pre- vs. post-LVAD changes. 

We hope our new tables clarify not only the content but also the frequency of themes across our respondent sample. 

4) Exclusion criteria should be added. 

We point out now in line 76 that no exclusion criteria were employed.

5) Please compare this research with the other similar results available in the literature.

While there is not much literature describing the subjective experiences of LVAD patients, we did add in lines 426-427 a reference to previous studies that qualitatively examined subjective changes resulting from LVAD treatment, as well as to a recent (2019) study of narrative medicine applications in helping LVAD patients during rehabilitation (see line 466).

6)      Please add limitations of the study using separate limitations section.

We appreciate the reviewers’ suggestion to draw attention to the study’s limitations. We do candidly address the primary limitations of our research – and of research into narratives in general – in the context of the “Function and Use of Narratives in the Clinical Setting” in our discussion section. Rather than extract these reflections into a separate section, we chose to keep these limitations here (and to elaborate upon them, referencing criticisms of the use of narratives that were recommended for inclusion in our paper by Reviewer 1), primarily because we feel that the limitations of our study belong within a larger discussion of limitations of the use of narratives in clinical settings. We hope that this reviewer will agree that pulling these limitations out of this section would decontextualize them from the larger literature on limitations of narrative inquiry in medicine.

Round  2

Reviewer 1 Report

I don't think that the authors have actually addressed the comments I made. In fact their response makes me worry even more that the understanding of narrative is very naîve.

The levels of similarity in intersubjective coding does not in and of itself mean that the data have not been framed in a particular way. Critiques of narrative show that patterns of narrative are often socially and culturally shared -- this is what a genre of narrative does, and my guess would be that the coders have a shared genre of narrative in mind, that is a shared expectation of the shape of these narratives. In addition, these narratives have emerged from data, but the data were gathered through semi-structured interview. The interview tool is not neutral and plays a strong shaping role in what interviewees say. For example, depending on how the interviews were done, they would have affected where research participants located the beginning of their story (how did they start, with which events), and this strongly shapes the arc of the story. The issues I was highlighting are related to subjective bias, but go beyond it.  The only reference now made to a potential negative view on narrative is Crispin Sartwell's 'End of Story', but this is not actually a book about narrative, it's about language, and it's about a prioritisation of language in a certain kind of philosophy. It is not the right reference here. A much better one, that is much more to the point, is Galen Strawson's Against Narrativity (which is not about medical narratives either, but it is much more relevant, and is discussed in, for example, the Woods paper). However, if the authors had read the introduction to Sartwell's End of Story carefully, they would have seen some of the issues that arise around narrative, and how these issues are also social, ethical and political.  I would strongly suggest at least another careful reading of Woods (who is not the last word on narrative in medicine, but at least presents some of the problems with a facile recourse to narrative in medicine). Narrative is a powerful tool, and it needs to be used with care and self-reflection. This is what I'm missing in this paper.

Author Response

Response to Reviewers 

Below we would like to address Reviewer 1’s comments on our resubmission. As Reviewers 2 and 3 said that we sufficiently addressed their concerns, we focus here on Reviewer 1’s concern:

I don't think that the authors have actually addressed the comments I made. In fact their response makes me worry even more that the understanding of narrative is very naîve. 

The levels of similarity in intersubjective coding does not in and of itself mean that the data have not been framed in a particular way. Critiques of narrative show that patterns of narrative are often socially and culturally shared -- this is what a genre of narrative does, and my guess would be that the coders have a shared genre of narrative in mind, that is a shared expectation of the shape of these narratives. In addition, these narratives have emerged from data, but the data were gathered through semi-structured interview. The interview tool is not neutral and plays a strong shaping role in what interviewees say. For example, depending on how the interviews were done, they would have affected where research participants located the beginning of their story (how did they start, with which events), and this strongly shapes the arc of the story. The issues I was highlighting are related to subjective bias, but go beyond it.  The only reference now made to a potential negative view on narrative is Crispin Sartwell's 'End of Story', but this is not actually a book about narrative, it's about language, and it's about a prioritisation of language in a certain kind of philosophy. It is not the right reference here. A much better one, that is much more to the point, is Galen Strawson's Against Narrativity (which is not about medical narratives either, but it is much more relevant, and is discussed in, for example, the Woods paper). However, if the authors had read the introduction to Sartwell's End of Story carefully, they would have seen some of the issues that arise around narrative, and how these issues are also social, ethical and political.  I would strongly suggest at least another careful reading of Woods (who is not the last word on narrative in medicine, but at least presents some of the problems with a facile recourse to narrative in medicine). Narrative is a powerful tool, and it needs to be used with care and self-reflection. This is what I'm missing in this paper. 

We appreciate the author’s insistence that narrative be acknowledged as a powerful tool that should be used with care and self-reflection. We would like to point out that we did actually read Sartwell’s, Strawson’s and Woods’ papers, respectively, very carefully and in an earnest attempt to see what we might be missing in our paper regarding a fair treatment of narratives as a whole, and of the significance of our narratives, specifically. 

My reading of Strawson’s article is that she argues against two main claims: 1) that self expression through narrative is fundamentally natural and healthy (a descriptive claim), and 2) that we ought to live our lives narratively because it is desirable for sense-making, particularly in the case of illness (a proscriptive claim).  We do not delve into Strawson’s work extensively in our paper because we do not make (or necessarily agree with) either of these claims. We merely make the observation that narrative can be and is viewed as useful for many individuals making a decision about whether or not to get an LVAD (please see our citation to this effect). This is an empirical observation based on feedback from the trial in which we tested the efficacy of our DA for increasing patient knowledge. Supplementary evidence from that trial also showed that patients and caregivers preferred the narrative part of our DA over other parts, because it helped them to better know what to expect after getting an LVAD. 

We do take the reviewer’s critique seriously and have extended our discussion of these limitations beginning in line 409. The new section hopefully satisfies some of the reviewers’ main concerns and alerts clinicians and patients to the potential limitations of narrative more generally.  We thank the reviewer for urging us to review this important literature.

Reviewer 2 Report

The authors have addressed my comments and the addition of tables is extremely helpful.

Author Response

We do not have any further responses to this reviewer, as he or she did not have any further critiques. Thank you for reviewing our manuscript.

Reviewer 3 Report

The authors have provided acceptable detailed responses to reviewers' critiques and have appropriately revised the manuscript. 

Author Response

(The authors gave the same response as above.)
